# National Study of Fournier Gangrene in Spain (2016–2021): Gender/Sex Differences in Mortality and Risks

**DOI:** 10.3390/medicina60101600

**Published:** 2024-09-29

**Authors:** Isabel Belinchón-Romer, Alberto Ramos-Belinchón, Ester Lobato-Martínez, Verónica Sánchez-García, José-Manuel Ramos-Rincón

**Affiliations:** 1Department of Clinical Medicine, Miguel Hernández University, 03550 Sant Joan d’Alacant, Spain; ibelinchon@umh.es; 2Department of Dermatology, Dr. Balmis General University Hospital, Alicante Institute for Health and Biomedical Research (ISABIAL), 03010 Alicante, Spain; sanchez_veronicagar@gva.es; 3Urology Department, Gregorio Marañón University Hospital, 28007 Madrid, Spain; albertoramosbel@gmail.com; 4Department of Internal Medicine, Dr. Balmis General University Hospital, Alicante Institute for Health and Biomedical Research (ISABIAL), 03010 Alicante, Spain; lobato_est@gva.es

**Keywords:** Fournier gangrene, hospitalizations, epidemiology, differences by gender, hospital mortality, Spain, comorbidity

## Abstract

*Background and Objective*: The aim of this study was to analyze sex differences in risk factors associated with hospital mortality in patients with Fournier gangrene (FG). *Materials and Methods:* A retrospective population-based study (2016–2021) included FG hospitalizations in Spain. To identify the risk factors, we used multivariable logistic regression and reported adjusted odds ratios (aORs) with 95% confidence intervals (CIs). *Results*: There were 3644 admissions for FG during the study period (82.5% men and 17.5% women). The mean hospitalization rate per 1000 admissions/year was 0.23 for men and 0.05 for women, and the mean hospitalization rate per 100,000 inhabitants/year was 2.7 for men and 0.4 in women. The most common comorbidities were hypertension (44.9%) and diabetes mellitus (35.6%). The main complications were sepsis (22.1%), 29.8% were admitted to the intensive care unit and 16.1% died. Mortality was higher in women (aOR 1.32, 95% CI 1.07–1.63). The main independent risk factors for mortality in the entire sample were older age, neoplasms, chronic kidney disease, heart failure, sepsis, acute kidney injury, and admission to the intensive care unit. In women, they were older age, leukemia, sepsis, acute kidney injury, and admission to the intensive care unit. *Conclusions:* The overall FG mortality rate was slightly higher in women than in men, people aged >64 years, people with chronic kidney disease, sepsis, and acute kidney injury, and admission to the intensive care unit. The independent factors associated with mortality in women were similar in both sexes.

## 1. Introduction

Jean Fournier first described the clinical characteristics of Fournier gangrene (FG) in 1883 based on five cases with no apparent cause [1]. FG is an acute necrotizing infection of the scrotum, penis, and perineum that can spread to the inguinal region, lower limbs, anterior abdominal wall, and even the chest [2,3]. It involves thrombosis of subcutaneous blood vessels secondary to obliterating endarteritis, causing tissue hypoxia and reduced vascular supply, which in turn facilitates the overgrowth of aerobic and anaerobic bacteria [4].

FG mainly affects men aged 50 to 70 years [5,6], and has a global incidence of 1.6 cases per 100,000 men/year [6]. Patients usually have underlying diseases such as diabetes, human immunodeficiency virus (HIV) infection, malignant neoplasms, or alcoholism, which lead to vascular and immune disorders that increase susceptibility to polymicrobial acute necrotizing infection [2,3,4,6]. FG has a high mortality rate ranging from 15% to 30% in the most recent series [2,3,4,6,7,8,9]. There are several studies that have explored epidemiological and prognostic factors in FG and [2,3,4,6] comparing men and women with the disease. Several scales are available to assess mortality risk (e.g., Fournier Gangrene Severity Index, Uludag Fournier Gangrene Severity Index, Simplified FGSI, and the Acute Physiology and Chronic Health Evaluation II scoring system [7,8]), and numerous studies have used these scales in different populations [9,10,11].

The Spanish Health Ministry maintains a compulsory Registry of Specialist Care Activities, which includes the demographic characteristics of hospitalized patients, called the minimum basic dataset [12]. Researchers have used this database to analyze the epidemiological and clinical aspects of numerous clinical entities, including urological and dermatological entities [13]. It is a valuable resource for estimating current case numbers and temporal trends [14].

Since the available evidence on FG risk factors is based on case series [2,3,4,6,7,8,9,10,11] and in the absence of in-depth research comparing men and women with the disease, we aim to provide a different perspective. Using administrative data from the Registry of Specialist Care Activities—Minimum Basic Dataset, we conducted a population-based study of patients admitted to the hospital with FG and analyzed sex differences in epidemiology and risk factors associated with hospital mortality

## 2. Materials and Methods

### 2.1. Study Design, Data Source, and Study Period

We conducted a retrospective observational study of patients with a diagnosis of FG discharged from a public hospital in Spain. We collected data from the Registry of Specialist Care Activities—Minimum Basic Dataset, which included patients’ demographic characteristics, as well as administrative and clinical variables related to the diagnoses and procedures performed during hospitalization, coded according to the International Classification of Diseases, 10th edition (ICD-10) [12]. The Spanish Health Ministry conducted periodic audits to evaluate registry accuracy.

We obtained data for people hospitalized with FG by searching for the relevant ICD-10 codes: N49.3 (FG) for men and N76.89 (other specified inflammation of vagina and vulva) for women. The ICD-10 code for men is specific to FG, while the code for women is not as specific. Since patients with FG typically require debridement, we excluded those who did not undergo surgery. To minimize the non-specific nature of the ICD code for women, we included only patients who went to the operating room. We identified 4755 cases of FG (3855 with the male-specific code and 890 with the female-specific code). Excluded from the study were 1101 patients (23.2% of the total), including 850 men (22.0%) and 251 women (22.8%)

### 2.2. Variables

We analyzed age, sex, length of hospital stay, comorbidities directly or indirectly related to FG, and prognostic factors (hypertension, diabetes mellitus, dyslipidemia, chronic kidney disease (CKD), obesity, chronic obstructive pulmonary disease (COPD), ischemic heart disease, heart failure, neoplasms, lymphoma, leukemia, HIV or hepatitis C infection, organ transplant), complications (sepsis, acute kidney injury (AKI), pulmonary embolism), type of discharge (home/deceased), and surgical intervention. Table 1 shows the ICD-10 codes of all clinical entities and comorbidities included in the study.

### 2.3. Data Analysis

The rate of hospitalization for FG was defined as the number of FG admissions per 100,000 inhabitants/year, obtained from the Spanish Statistics Institute (https://www.ine.es/en/, accessed on 26 September 2024), and the number of FG admissions per 1000 all-cause admission years, obtained from the Registry of Specialist Care Activities—Minimum Basic Dataset (https://pestadistico.inteligenciadegestion.sanidad.gob.es/publicoSNS/N/rae-cmbd/rae-cmbd, accessed on 26 September 2024). Hospitalization and mortality rates were adjusted by age using the direct method of standardization, with Eurostat 2013 data as reference population

Categorical variables were presented as absolute values and percentages. For continuous variables, we calculated means and standard deviations if the data were normally distributed according to the Kolmogorov–Smirnov test and medians and interquartile ranges (IQRs) if they were not.

To analyze the differences in hospital mortality between men and women, we calculated odds ratios (ORs) with 95% confidence intervals (CIs).

Categorical variables were analyzed using the chi-square test or Fisher’s exact test, and continuous variables were analyzed using the Mann–Whitney U test, because the distribution was not normal (*p* value < 0.05 in Kolmogorov–Smirnov test). A *p* significance was set as significant. We used multivariable logistic regression analysis to identify independent predictors of in-hospital mortality. Significant variables in the univariate analysis were entered into a multivariate logistic regression model using a stepwise selection method with the likelihood ratio test. The CoxSnell R2 statistic and the Nagelkerke R2 statistic were used to measure the strength of the relationship between dependent (mortality) and independent variables. To test the model fit, we calculated the area under the receiver operating characteristic curve (AUC) with a 95% CI. IBM SPSS for Windows (version 25.0; IBM Corp., Armonk, NY, USA) was used for all the statistical analyses.

### 2.4. Ethical Aspects

This study did not require ethics committee approval or informed consent from participants, according to Spanish legislation. To guarantee patient anonymity, the Spanish Health Ministry provided a database after removing all possible patient identifiers.

## 3. Results

### 3.1. Yearly Evolution

During the study period, there were 26.4 million hospital admissions, of which 3644 were for FG. The number of annual FG admissions increased from 457 in 2016 to 781 in 2021. The number of yearly admissions was more than four times greater in men (82.5%) than in women (17.5%) (Figure 1). The mean hospitalization rates were 0.14 per 1000 admissions/year (0.23 for men and 0.05 for women) and 1.7 per 100,000 inhabitants/year (2.7 for men and 0.4 in women). The mean hospitalization rate per 1000 admissions/year was 0.23 for men and 0.05 for women. The crude and standardized hospitalization rates were approximately five times higher in men than in women (Table 2).

### 3.2. Epidemiological Characteristics of Participants

Of the 3644 FG admissions, 82.5% were men and 17.5% were women The main comorbidities (in descending order) were hypertension, diabetes mellitus, dyslipidemia, smoking, obesity, and neoplasms. The main complications were sepsis and AKI. About 30% of cases involved admission to the intensive care unit (ICU) for a median duration of 5 days. The reason for discharge was death in almost one-fifth of cases. The median length of the hospital stay was 18 days (Table 3).

### 3.3. Sex Differences in Participant Characteristics

Table 3 shows sex differences in age, comorbidities, complications, and FG course. The main epidemiological difference was in the proportion of women versus men aged > 80 years (*p <* 0.001). Regarding comorbidities, women had a lower prevalence of smoking, alcohol consumption, neoplasms, COPD, ischemic heart disease, and cirrhosis, but a higher prevalence of obesity, CKD, and heart failure (*p* < 0.05). The median length of hospital stay was shorter in women (*p* = 0.002). The proportion of admission to ICU and days of admission in ICU, complications during hospitalization, and the types of infections diagnosed during hospitalization were similar in both sexes.

### 3.4. Crude and Standarized Rate of Mortality

The mortality rate was higher in women than in men (20.9% vs. 15.1% with OR 1.31, 95% CI 1.09–1.56). The crude and standardized rate of mortality is in Table 4.

### 3.5. Sex Differences in Mortality

We also found higher mortality odds rates in women than in men when we analyzed people with hypertension, diabetes, those admitted to the ICU, and those aged 65 to 79 years. We also found a higher mortality odds rate in women than in men when the patients had no other comorbidities and complications. There were no differences in the other variables analyzed (Table 5).

### 3.6. Risk Factors for Mortality among All Patients

Table 6 shows the risk factors for mortality in men, women, and both, and in Table 7 the crude ORs are shown. Figure 2 presents the adjusted ORs (aORs) obtained through multivariable analysis. The adjusted mortality risk was higher in women (aOR 1.32, 95% CI 1.07–1.63); people aged 65–79 years (aOR 2.09, 95% CI 1.70–2.56) and 80 years or older (aOR 5.92, 95% CI 4.66–7.53); people with neoplasms (aOR 3.97, 95% CI 3.22–4.90), CKD (aOR 1.77, 95% CI 1.236–2.30), heart failure (aOR:136, 95%CI 1.04–1.79), leukemia (aOR:2.39 95%CI: 1.01–5.19), sepsis as an acute complication (aOR 3.53, 95% CI 2.88–4.31), and AKI (aOR 2.42, 95% CI 1.96–2.99); and people admitted to the ICU (aOR 1.55, 95% CI 1.27–1.89). The model has a Cox–Snell R^2^ value of 0.226 and a Nagelkerle R^2^ value of 0.352. The AUC of the model was 0.831 (95% CI 0.817–0.844, *p <* 0.001).

### 3.7. Risk Factors for Mortality among Men and Women

Similar variables were associated with a higher mortality risk in men as in the whole study sample: age 65–79 years, age 80 years or older, neoplasms, CKD, heart failure, sepsis, AKI, and ICU admission (Figure 2). The model has a Cox–Snell R^2^ value of 0.184 and a Nagelkerle R^2^ value of 0.321. The AUC of the model was 0.841 (95% CI 0.824–0.859, *p <* 0.001). The factors associated with higher mortality in women were age 65–79 years, age ≥ 80 years, leukemia, sepsis, AKI, and ICU admission. The model has a Cox–Snell R^2^ value of 0.213 and a Nagelkerle R^2^ value of 0.333. The AUC of the model was 0.827 (95% CI 0.790–0.863 *p <* 0.001).

## 4. Discussion

This study described the hospitalization rates of men and women with FG in Spain. We observed a high mortality rate in both sexes, although it was slightly higher in women than that in men. There were some minor differences between the sexes in disease epidemiology and other characteristics, with fewer comorbidities in women. Mortality was associated with older age, and was particularly high in people aged 80 years. Other factors associated with mortality were underlying neoplasms (both sexes), CKD (only men), sepsis as a complication (both sexes), AKI (both sexes), ICU admission (only men), and not undergoing surgery (only men).

In a 2009 study from the USA by Sonrensen et al., the rate of FG hospitalization was 1.6 cases per 100,000 men/year, which represented 0.02% of all hospital admissions in men [5]. We found a slightly higher hospitalization rate, with 2.7 cases per 100,000 men/year, or 0.023% of all hospital admissions in men/year. The differences between the two studies may be attributed to the different epidemiological and social situations in the populations. In women, we found an FG hospitalization rate of 0.64 per 100,000 women/year or 0.005% of all admissions. We cannot compare these results with previous findings because we identified no studies that specifically reported FG hospitalization in women.

FG predominantly affects men [15,16]; in fact, some publications have reported no cases in women [17,18], although others have found a prevalence ranging from 13% to 30% [9,11,15,19]. The proportion of women observed in our study (17.5%) was within the upper range reported in the most recent publications, possibly because we searched for ICD-10 diagnostic code N76.89. However, there are an increasing number of reports on FG in women [9,11]. Different case series have shown higher mortality in women than in men with this condition [10,19,20,21], which is in line with our findings.

According to previous studies, the main comorbidities associated with FG include diabetes, alcoholism, cardiovascular disease, and alcohol consumption [11,15,16,17]. The main comorbidities in our study were hypertension (44.9%), diabetes mellitus (35.3%), dyslipidemia (25.3%), smoking (18.3%), obesity (14.3%), and neoplasms (12.6%), and when we analyzed the differences by gender/sex, a smaller proportion of women had hypertension, diabetes mellitus, or a smoking habit and a larger proportion were obese.

In our population-based study, older age was the primary risk factor for mortality, as other studies [22,23]. In contrast, other studies have shown no association between older age alone and mortality risk [11,24].

Several meta-analyses have reported an association between kidney failure and increased mortality [9,25,26,27]. In our study, CKD and AKI were the risk factors for mortality. Early diagnosis of acute injury and consideration of solutions such as dialysis may provide patients with a better chance of survival [9].

Men and women with neoplasms had a higher mortality risk in our study, as reported in previous publications [9,27,28]. Several studies have linked hypotension with mortality [11,27]. Although we were unable to specifically evaluate hypotension, we analyzed sepsis accompanied by hypotension and AKI. Sepsis and AKI were relevant mortality risk factors in both men and women with FG. Therefore, as other authors have commented [11,28,29], it is important to diagnose hemodynamic instability and parameters of sepsis and septic shock in people with infections, specifically FG, to intensify therapeutic efforts [28,29].

Previous studies have linked ICU admission to higher mortality [11,30]. Individuals who require ICU admission typically have hemodynamic instability related to hypotension, sepsis, and kidney failure. The multivariable analysis we fitted to define the confounding factors showed significant associations between mortality and these variables, including sepsis, acute kidney injury, and ICU admission, and independent factors of poor prognosis [31].

In this study, we were unable to obtain information regarding treatment. The therapeutic strategy for FG should be based on hemodynamic stabilization, early radical surgical debridement, wide-spectrum antibiotic therapy, daily cleaning and dressing, hyperbaric oxygen therapy in some cases, and hyperbaric oxygen therapy [15,29]. Initial debridement with sufficient resection of nonviable tissues is considered the most important factor for survival [25,29]. In our study, we include patients who have undergone surgical debridement. However, we were unable to evaluate the number of debridements each patient received. As we had no information on non-surgical treatments, we were unable to evaluate their impact on mortality.

The main strength of this study was that our data source enabled the analysis of FG hospitalization and mortality risk factors for each sex separately. While other studies have analyzed mortality risk factors in patients with FG, none have included an in-depth comparison of the findings in men and women.

However, our study had several limitations. First, it relies on administrative databases, which assumes that the ICD-10 codes used accurately represent the same disease (one code is specific to FG in men, while the code for women is not specific to FG). Since patients with FG typically require debridement, we included only those who underwent surgery to reduce the nonspecific nature of the ICD code for women. Second, the database recorded admissions rather than patients, and if a patient was admitted more than once for FG, each admission was treated as a separate unit of analysis. Third, the information available for each admission included a set of coded diagnoses at discharge. Attending clinicians sometimes record only the diagnoses that they consider most relevant, and do not always use diagnostic codes. Consequently, we may have been missed. Fourth we had no information on the extent of FG (perineal, abdominal, etc.) or non-surgical treatment (pharmacotherapy, fluid therapy, etc.). Fifth, this type of administrative database contained no laboratory results, meaning that we were unable to analyze the different FG mortality risk scales. Sixth, we did not have detailed microbiological information and only generic ICD-10 codes for microbiological infections. Finally, the time from symptom onset to admission is unknown. This is an important factor because people who die from FG tend to have a longer duration of symptoms at admission [9].

## 5. Conclusions

In conclusion, FG predominantly affected men with approximately 2 out of every 10 cases occurring in women. We observed a high FG mortality rate, which was slightly higher in women, people aged ≥ 65 years (especially those aged ≥ 80 years), and people with CKD, sepsis, AKI, and ICU admission. The independent factors associated with mortality were similar in both sexes, except for neoplasia and CKD (not significant in women). The Spanish medical administrative registry provides key information about the public health burden of FG in Spain, highlighting comorbidity and mortality in both sexes.

## Figures and Tables

**Figure 1 medicina-60-01600-f001:**
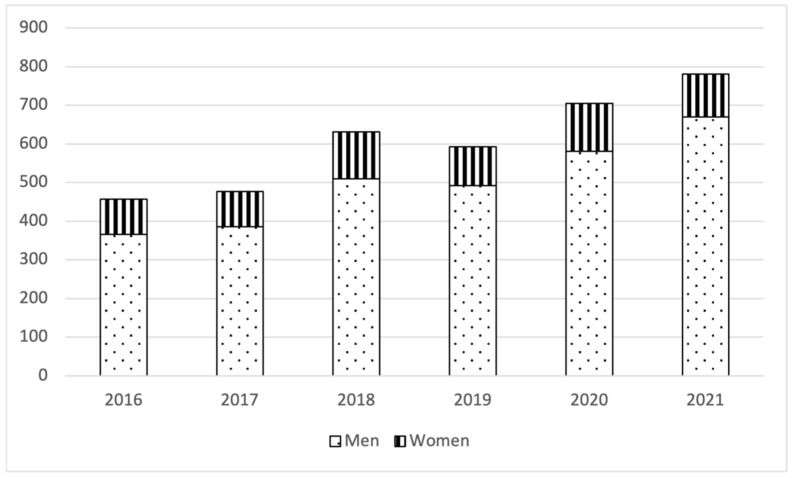
Yearly Fournier gangrene, in all patients, men and women.

**Figure 2 medicina-60-01600-f002:**
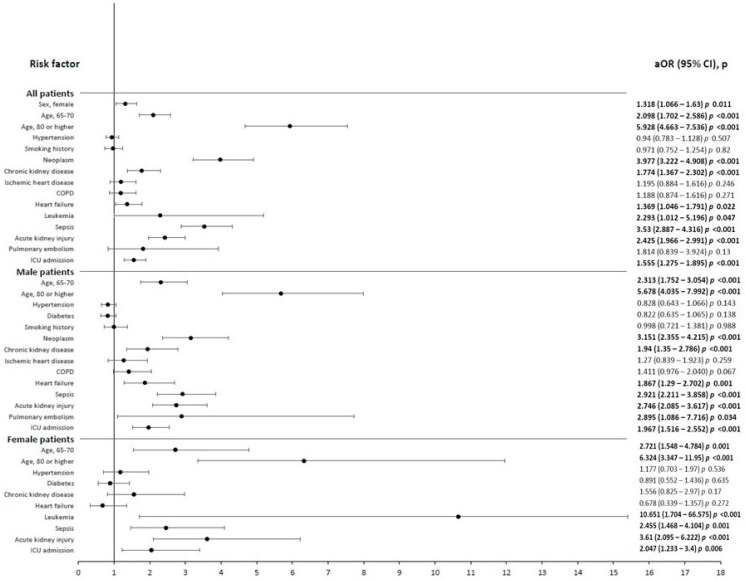
Independent risk factors for hospital mortality in all patients, men and women with Fournier gangrene (adjusted odds ratios with 95% confidence intervals).

**Table 1 medicina-60-01600-t001:** International Classification Disease 10th edition (ICD-10) codes used in the study.

	ICD-10 Codes
**Diseases**	
Fournier Gangrene for men	N49.3
Fournier gangrene in women (other specified inflammation of vagina and vulva)	N76.89
Comorbidities	
Hypertension	I10; I11; I12; I13; I15
Diabetes mellitus	E10; E11
Dyslipidemia	E78
Smoking	F17
Neoplasms	C01; C02; C03; C04; C05; C06; C07; C08; C09; C10; C11; C12; C13; C14; C15; C16; C17; C18; C19; C20; C21; C22; C23; C24; C25; C26; C27; C28; C29; C30; C31; C32; C33; C34; C35; C36; C37; C38; C39; C40; C41; C42; C43; C44; C45; C46; C47; C48; C49; C50; C51; C52; C53; C54; C55; C56; C57; C58; C59; C60; C61; C62; C63; C64; C65; C66; C67; C68; C69; C70; C71; C72; C73; C74; C75; C76; C77; C78; C79; C80
Obesity	E66
Chronic kidney disease	N18
Alcohol consumption	F10
Ischemic heart disease	I21; I22; I23; I24; I25
COPD	J44
Heart failure	I50
Leukemia	C92; C93; C94; C95; C96
Lymphoma	C81; C82; C83; C84; C85; C86; C87; C88; C89; C90; C91
Cirrhosis	K74
Vascular disease	I73
Hepatitis C	B18.2; B19.20; B19.21
HIV	B20
Transplant	Z94
Complications	
Sepsis	A40; A41
Acute kidney injury	N17
Pulmonary embolism	I26
Microbiology	
*Escherichia coli*	B96.2
*Streptococcus*	B95.0; B95.1; B95.4; B95.5
*Staphylococcus*	B96.6; B95.7; B95.8
*Pseudomonas*	B96.5

**Table 2 medicina-60-01600-t002:** Episodes and crude and standardized rate of hospitalization per 1000 all-cause admissions and per 100,000 inhabitants (men, women, and both).

	Men			Women			Total		
	*N*	RH ^a^	RH ^b^	*N*	RH ^a^	RH ^b^	*N*	RH ^a^	RH ^b^
	**Crude Rates**								
2016	366	0.17	1.6	91	0.04	0.4	457	0.10	1.0
2017	386	0.18	1.7	91	0.04	0.4	477	0.10	1.0
2018	510	0.23	2.2	121	0.05	0–5	631	0.14	1.4
2019	492	0.22	2.1	101	0.04	0.4	593	0.13	1.3
2020	581	0.30	2.5	124	0.06	0.5	705	0.17	1.5
2021	670	0–32	2.9	111	0.05	0.5	781	0.18	1.6
Mean	490	0.23	2.7	106	0.05	0.4	596	0.14	1.7
	Standarized rates *								
2016	-	0.182	1.7	-	0.042	0.4	-	0.107	1.0
2017	-	0.186	1.8	-	0.039	0.4	-	0.108	1.0
2018	-	0.239	2.4	-	0.049	0.5	-	0.138	1.4
2019	-	0.231	2.3	-	0.041	0.4	-	0.131	1.3
2020	-	0.319	2.6	-	0.061	0.5	-	0.182	1.5
2021	-	0.330	3.0	-	0.047	0.4	-	0.181	1.6
Mean	-	0.24	3.0	-	0.05	0.4	-	0.14	1.3

* Adjusted by age using the direct method of standardization. Abbreviations: *N*, sample size (number of admissions); RH, rate of hospitalizations. ^a^ Number of hospitalizations for Fournier gangrene per 1000 all-cause admissions per year. ^b^ Number of hospitalizations for Fournier gangrene per 100,000 inhabitants per year.

**Table 3 medicina-60-01600-t003:** General characteristics of the population and differences between men and women.

	Total ^a^	Men ^a^	Women ^a^	*p* Value ^b^
Total	3641 (100)	3004 (82.5)	637 (17.5)	—
Age (years)				
median (IQR)	63(49–76)	63 (55–73)	62 (49–76)	0.059
<64 y	1950 (53.5)	1605 (53.4)	345 (51.0)	**<0.001**
65–70 y	1179 (22.4)	1007 (33.5)	173 (26.9)	
≥80 y	515 (14.1)	293 (13.1)	122 (19.7)	
Primary diagnosis	3114 (85.5)	2618 (87.1)	496 (77.5)	**<0.001**
Comorbidities				
Hypertension	1637 (44.9)	1371 (45.6)	266 (41.6)	0.066
Diabetes mellitus	1288 (35.3)	1060 (35.3)	228 (35.7)	0.845
Dyslipidemia	921 (25.3)	761 (25.3)	160 (25.0)	0.920
Smoking	666 (18.3)	602 (20.0)	63 (10.0)	**<0.001**
Obesity	520 (14.3)	385 (12.8)	135 (21.1)	**<0.001**
Neoplasms	458 (12.6)	392 (13.0)	66 (10.3)	0.066
Alcohol consumption	344 (9.4)	335 (11.1)	9 (1.4)	**<0.001**
Chronic kidney disease	306 (8.4)	239 (7.9)	68 (10.6)	**0.028**
COPD	262 (7.2)	252 (8.4)	10 (1.6)	**<0.001**
Heart failure	258 (7.1)	195 (6.5)	63 (9.9)	**0.003**
Ischemic heart disease	229 (6.3)	206 (6.9)	23 (3.6)	**0.003**
Hepatitis C	55 (1.5)	50 (1.7)	5 (0.8)	0.109
Cirrhosis	37 (1.0)	36 (1.29	1 (0.2)	**0.014**
Lymphoma	35 (1.0)	24 (0.9)	11 (1.7)	**0.042**
Leukemia	27 (0.7)	20 (0.7)	7 (1.1)	0.250
Vascular disease				
Transplant	26 (0.7)	24 (0.8)	2 (0.39	0.185
HIV	18 (0.5)	15 (0.5)	3 (0.5)	1.00
Complications				
Sepsis	828 (22.7)	669 (23.3)	159 (24.9)	0.151
Acute kidney injury	627 (17.2)	16.9 (50)	120 (18.8)	0.249
Pulmonary embolism	29 (0.9)	23 (0.8)	6 (0.9)	0.625
ICU admission	189 (29.8)	1119 (31.1)	930 (31.4)	0.436
Days in ICU, median (IQR)	5 (2–11)	4 (2–11)	6 (2–13)	0.140
Hospital mortality	588 (20.9)	455 (15.1)	133 (20.9)	**0.002**
Days in hospital, median (IQR)	18 (8–33)	18 (9–33)	16 (6–32)	**0.002**
Surgical intervention				
<48 h after intervention	2918 (80.1)	2398 (79.8)	520 (81.4)	0.365
Microbiology (ICD-10 code)				
*Escherichia coli*	519 (14.2)	427 (14.29	92 (14.4)	0.902
*Streptococcus*	269 (7.4)	223 (7.4)	46 (7.29	0.845
*Staphylococcus*	190 (5.2)	166 (5.5)	24 (3.9)	0.068
*Pseudomonas*	163 (4.5)	140 (4.7)	23 (3.6)	0.239

^a^ The results are presented as number (%) of admissions unless otherwise specified. ^b^ Significant *p* values (<0.05) are in bold. Abbreviations: COPD, chronic obstructive pulmonary disease; HIV: human immunodeficiency virus; ICD-10, International Classification of Diseases, 10th Revision; ICU, intensive care unit, IQR, interquartile range; *N*, sample size (number of admissions).

**Table 4 medicina-60-01600-t004:** Episodes and crude and standardized rate * of mortality per 1000 Spanish cases of death and per 100,000 inhabitants (men, women, and both).

	Men			Women			Total		
	*N*	RH ^a^	RH ^b^	*N*	RH ^a^	RH ^b^	*N*	RH ^a^	RH ^b^
	Crude rates								
2016	67	0.22	0.29	19	0.08	0.08	86	0.15	0.18
2017	67	0.31	0.29	15	0.07	0.06	82	0.19	0.18
2018	65	0.30	0.28	25	0.19	0.10	90	0.21	0.19
2019	66	0.31	0.28	23	0.11	0.10	89	0.21	0.19
2020	87	0.35	0.37	25	0.10	0.10	112	0.23	0.23
2021	103	0.44	0.44	26	0.12	0.10	129	0.29	0.27
Mean	75	0.31	0.33	22	0.010	0.09	97	0.21	0.21
	Standardized rates *								
2016	-	0.4	0.33		0.33	0.08		0.43	0.19
2017	-	0.50	0.32		0.14	0.05		0.38	0.17
2018	-	0.41	0.31		0.24	0.09		0.35	0.19
2019	-	0.38	0.31		0.17	0.08		0.31	0.18
2020	-	0.63	0.40		0.48	0.10		0.58	0.24
2021	-	0.58	0.48		0.23	0.09		0.46	0.27
Mean	-	0.50	0.36		0.27	0.08		0.42	0. 21

* Adjusted by age using the direct method of standardization. Abbreviations: *N*, sample size (number of admissions); RH, rate of hospitalizations. ^a^ Number of hospitalizations for Fournier gangrene per 1000 all-cause deaths during admission per year. ^b^ Number of hospitalizations for Fournier gangrene per 100,000 inhabitants per year.

**Table 5 medicina-60-01600-t005:** Comparison of hospital mortality in men and women.

	HM in men, % (*n*/*N*) ^a^	HM in women, % (*n*/*N*) ^a^	OR (95% CI)	*p* Value ^b^
Total	15.1 (455/3004)	20.9 (133/637)	1.48 (1.19–1.83)	**0.001**
Age in years				
<64 y	7.9 (127/1605)	10.2 (35/344)	0.98 (0.69–1.38)	0.163
65–80 y	20.0 (201/1006)	28.7 (49/171)	1.44 (1.03–1.96)	**0.015**
≥80 y	32.3 (127/393)	40.2 (49/122)	1.29 (0.94–1.78)	0.126
Hypertension				
No	13.3 (218/1634)	14.5 (54/372)	1.03 (0.80–1.52)	0.557
Yes	17.3 (237/1370)	29.8 (79/265)	2.03 (1.50–2.73)	**<0.001**
Diabetes				
No	16.2 (315/1944)	18.2 (75/411)	1.15 (0.87–1.52)	0.311
Yes	13.2 (150/1060)	25.5 (58/226)	2.27 (1.60–3.21)	**<0.001**
Dyslipidemia				
No	15.1 (338/2244)	20.5 (98/477)	1.45 (1.13–1.87)	**0.003**
Yes	15.4 (117/760)	21.9 (35/160)	1.53 (1.00–2.15)	0.060
Smoking				
No	16.1 (387/2402)	21.6 (124/573)	1.14 (1.43–1.80)	**0.002**
Yes	11.3 (68/602)	14.1 (9/64)	1.28 (0.60–2.71)	0.532
Neoplasms				
No	12.8 (335/2612)	20.0 (114/571)	1.69 (1.34–2.14)	**<0.001**
Yes	30.6 (120/392)	28.8 (19/66)	0.91 (0.51–1.62)	0.885
Obesity				
No	15.3 (401/2620)	20.7 (104/503)	1.44 (1.13–1.83)	**0.003**
Yes	14.1 (54/384)	21.6 (29/134)	1.68 (1.02–2.78	0.055
Chronic kidney disease				
No	13.7 (378/2766)	18.1 (103/569)	1.39 (1.09–177)	**0.006**
Yes	32.4 (77/238)	44.1 (30/68)	1.65 (0.95–2.86)	0.084
Alcohol consumption				
No	14.9 (398/2669)	20.9 (131/628)	1.50 (1.21–1.87)	**<0.001**
Yes	17.0 (57/335)	22.2 (2/9)	1.39 (0.28–6.88)	0.656
Ischemic heart disease				
No	14.8 (413/2798)	20.5 (126/614)	1.49 (1.19–1.86)	**<0.001**
Yes	20.4 (42/206)	30.4 (7/23)	1.71 (0.66–4.42)	0.286
COPD				
No	14.3 (394/2752)	20.7 (130/627)	1.56 (1.26–1.95)	**<0.001**
Yes	24.2 (61/252)	30.0 (3/10)	1.34 (0.33–5.34)	0.710
Heart failure				
No	13.4 (377/2810)	19.2 (110/574)	1.53 (1.21–1.93)	**<0.001**
Yes	40.2 (78/194)	36.5 (23/63)	0.85 (0.76–1.53)	0.603
Leukemia				
No	15.0 (449/2984)	20.5 (129/630)	1.45 (1.17–1.80)	**<0.001**
Yes	30.0 (6/20)	57.1 (4/7)	3.11 (0.52–18.3)	0.376
Lymphoma				
No	15.0 (448/2980)	20.8 (130/626)	1.48 (1.19–1.84)	**<0.001**
Yes	29.2 (7/24)	27.3 (3/11)	0.91 (1.18–4.47)	1.00
Cirrhosis				
No	15.0 (446/2968)	20.9 (579/3604)	1.49 (1.20–1.85)	**<0.001**
Yes	25.0 (9/36)	0.0 (0/1)	NA	1.00
Hepatitis C				
No	15.0 (443/2954)	20.9 (132/632)	1.49 (1.20–1.85)	**<0.001**
Yes	24.4 (12/50)	20.0(1/5)	0.79 (0.08–7.78)	1.00
HIV				
No	15.2 (454/2989)	20.7 (131/634)	1.45 (1.17–1.80)	**0.001**
Yes	6.7 (1/15)	66.7 (2/3)	28.0 (1.2–648)	0.056
Transplant				
No	15.0 (448/2980)	20.9 (133/635)	1.49 (1.20–1.83)	**<0.001**
Yes	29.2 (7/24)	29.2 (0/2)	NA	1.00
Sepsis				
No	9.2 (214/2236)	13.4 (64/479)	1.53 (1.13–2.06)	**0.005**
Yes	36.1 (241/668)	43.7 (69/158)	1.37 (0.96–1.95)	0.083
Acute kidney injury				
No	9.9 (248/2497)	13.7 (71/518)	1.44 (1.08–1.91)	**0.011**
Yes	40.8 (207/507)	52.1 (62/119)	1.56 (1.05–2.35)	0.035
Pulmonary embolism				
No	15.9 (447/2981)	20.8 (131/631)	1.48 (1.19–1.84)	**<0.001**
Yes	34.8 (8/23)	33.3 (2/6)	0.93 (0.14–6.28)	1.00
ICU admission				
No	10.1 (205/2033)	15.1 (67/445)	1.58 (1.17–2.18)	**0.002**
Yes	26.4 (245/929)	34.8 (65/187)	1.48 (1.06–2.07)	**0.025**

^a^ The results are presented as percentage (number of events/sample size). ^b^ Significant *p* values (<0.05) are in bold. Abbreviations: NA: not available; CI, confidence interval; COPD, chronic obstructive pulmonary disease; HIV, human immunodeficiency virus; HM, hospital mortality; ICU, intensive care unit; NA, not applicable; OR, odds ratio.

**Table 6 medicina-60-01600-t006:** Risk factors for hospital mortality in all patients (*n* = 3004), men (n = 3004), and women (*n* = 558) with Fournier gangrene.

	All Patients		Men		Female	
	Hospital Mortality, % (*n*/*N*) ^a^	*p* Value ^b^	Hospital mortality, % (*n*/*N*) ^a^	*p* Value ^b^	Hospital mortality, % (*n*/*N*) ^a^	*p* Value ^b^
Total	16.1 (588/3641)	—	15.1 (445/3004)	—	20.9 (133/558)	—
Sex						
Male	15.1 (445/3004)		—	—	—	—
Female	20.9 (133/558)	**<0.001**	—	—	—	—
Age						
< 64 y	8.3 (162(1949)	—	7.9 (127/1605)	—	10.2 (35/344)	—
65–70 y	21.2 (250/1177)	**<0.001**	20.0 (201/1006)	**<0.001**	28.7 (49/171)	**<0.001**
≥ 80 y	34.0 (176/517)	**<0.001**	32.3 (127/393)	**<0.001**	40.2 (49/122)	**<0.001**
Hypertension						
No	13.6 (272/2006)	—	13.3 (218/634)	—	14.5 (54/372)	—
Yes	19.3 (316/163)	**<0.001**	17.3 (237–1370)	**0.003**	29.8 (79/265)	**<0.001**
Diabetes						
No	16.6 (390/2355)	—	16.2 (315/1944)	—	18.2 (75/411)	—
Yes	15.4 (198/1288)	0.371	13.2 (140/1060)	**0.029**	**25.7 (58/226)**	**0.032**
Dyslipidemia						
No	16.0 (436/2722)	—	15.1 (338/2244)	—	20.5 (98/477)	—
Yes	16.5 (152/921)	0.717	15.4 (117/760)	0.398	21.9 (35/160)	0.720
Smoking						
No	17.2 (511/2977)	—	16.1 (387/2402)	—	21.0 (124/573)	—
Yes	11.6 (77/666)	**<0.001**	11.3 (68/602)	**0.003**	14.1 (9/64)	0.195
Neoplasms						
No	14.1 (449/3185)	—	12.8 (335/2612)	—	20.0 (114/571)	—
Yes	30.3 (139/458)	**<0.001**	30.6 (120/392)	**<0.001**	28.8 (19/66)	0.916
Obesity						
No	16.2 (505/3124)	—	15.3 (401/2620)	—	20.7 (104/503)	—
Yes	16.0 (83/519)	0.946	14.1 (54/384)	0.593	21.6 (29/134)	0.811
Chronic kidney disease						
No	14.4 (481/335)	—	13.7 (378/2766)	—	18.1 (103/569)	—
Yes	34.7 (107/308)	**<0.001**	32.4 (77/238)	**<0.001**	44.1 (30/68)	**<0.001**
Alcohol consumption						
No	16.0 (529/3299)	—	14.9 (398/2669)	—	20.9 (131/628)	—
Yes	17.2 (59/344)	0.592	17.5 (57/335)	0.332	22.2 (2/9)	0.988
Ischemic heart disease						
No	15.8 (539/3414)	—	14.8 (413/2798)	—	20.8 (126/614)	—
Yes	21.4 (49/229)	**0.032**	20.4 (42/206)	**0.034**	30.4 (7/23)	0.293
COPD						
No	15.5 (524/3381)		14.3 (394/2752)	—	20.7 (130/627)	—
Yes	24.4 (64/262)	**<0.001**	24.2 (61/252)	**<0.001**	30.0 (3/10)	0.443
Heart failure						
No	14.4 (487/3386)	—	13.4 (377/2810)	—	19.2 (110/574)	—
Yes	39.3 (101/257)	**<0.001**	40.2 (78/194)	**<0.001**	26.5 (23/63)	**0.003**
Leukemia						
No	16.0 (578/3616)	—	15.0 (449/2984)	—	20.5 (129/630)	—
Yes	37.0 (10/17)	**0.007**	30.0 (6/20)	0.106	57.1 (4/7)	**0.038**
Lymphoma						
No	16.0 (578/3608)	—	15.0 (4482980)	—	20.3 (130/627)	—
Yes	28.6 (10/35)	0.061	29.2 (7/24)	0.078	27.3 (3711)	0.706
Cirrhosis						
No	16.1 (579/2606)	—	15.0 (446/2968)	—	20.9 (133/636)	—
Yes	24.3 (9/37)	0.17	25.0 (9/36)	0.102	0.0 (0/11)	1.00
Hepatitis C						
No	16.0 (575/3588)	—	15.3 (443/2954)	—	20.9 (132/632)	—
Yes	23.6 (13/55)	0.138	24.0 (12/60)	0.531	20.0 (1/5)	1.00
HIV						
No	16.1 (585/3625)	—	15.2 (454/2989)	—	20.7 (131/634)	—
Yes	16.7 (3/18)	1.00	6.7 (1/15)	0.213	66.7 (273)	0.112
Transplant						
No	16.1 (581/3617)	—	15.0 (448/2980)	—	20.9 (133/635	—
Yes	26.9 (7/26)	0.174	29.2 (7/24)	0.078	0.0 (0/2)	1.00
Sepsis						
No	9.9 (278/2817)	—	9.2 (214/2336)	—	13.4 (64/479)	—
Yes	37.5 (310/826)	**<0.001**	36.1 (241/668)	**<0.001**	43.7 (69/479)	**<0.001**
Acute kidney injury						
No	10.6 (319/3017)	—	9.9 (248/2497)	—	13.7 (71/518)	—
Yes	43.0 (269/626)	**<0.001**	40.8 (207/507)	**<0.001**	**52.1 (62/119)**	**<0.001**
Pulmonary embolism						
No	16.0 (578/3614)	—	15.0 (447/2981)	—	20.8 (131/631)	—
Yes	34.5 (10/29)	**0.018**	34.8 (8/23)	**0.016**	**33.3 (2/6)**	0.610
ICU admission						
No	10.7 (272/2480)	—	10.1 (205/2033)	—	15.3 (67/445)	—
Yes	27.8 (310/1160)	**<0.001**	26.4 (245/929)	**<0.001**	**34.8 (65/187)**	**<0.001**

^a^ The results are presented as percentage (number of events/sample size). ^b^ Significant *p* values (<0.05) are in bold. Abbreviations: COPD, chronic obstructive pulmonary disease; HIV, human immunodeficiency virus; ICU, intensive care unit; *N*, sample size (number of admissions).

**Table 7 medicina-60-01600-t007:** Odds ratio for hospital mortality in all patients (*n* = 3004), men (*n* = 3004), and women (*n* = 558) with Fournier gangrene.

	All Patients		Men		Female	
	OR (95% CI)	*p* Value ^a^	OR (95% CI)	*p* Value ^a^	OR (95% CI)	*p* Value ^a^
Total	—	—	—	—	—	—
Sex						
Male	1		—	—	—	—
Female	1.48 (1.19–1.83)	**<0.001**	—	—	—	—
Age						
<64 y	1	—	1	—	1	—
65–70 y	2.98 (2.40–3.68)	**<0.001**	2.90 (2.30–3.70)	**<0.001**	3.54 (2.19–5.74)	**<0.001**
≥80 y	5.69 (4.40–7.35)	**<0.001**	5.55 (4.20–7.34)	**<0.001**	5.92 (3.8–9.80)	**<0.001**
Hypertension						
No	1	—	1	—	1	—
Yes	1.55 (1.27–1.82)	**<0.001**	1.35 (1.11–1.65)	**0.003**	2.50 (1.69–3.62)	**<0.001**
Diabetes						
No	1	—	1	—	1	—
Yes	0.91 (0.76–1.10)	0.371	0.78 (0.63–0.97)	**0.029**	1.54 (1.05–2.29)	**0.032**
Dyslipidemia						
No	1	—	1	—	1	—
Yes	1.03 (0.84–1.26)	0.717	1.02 (0.82–1.29)	0.398	1.08 (0.70–1.64)	0.720
Smoking						
No	1	—	1	—	1	—
Yes	0.63 (0.49–0.81)	**<0.001**	0.66 (0.50–0.87)	**0.003**	0.59 (0.28–1.23)	0.195
Neoplasms						
No	1	—	1	—	1	—
Yes	2.65 (2.12–3.21)	**<0.001**	2.99 (2.35–3.82)	**<0.001**	1.62 (0.91–2.86)	0.916
Obesity						
No	1	—	1	—	1	—
Yes	0.98 (0.76–1.27)	0.946	0.90 (0.67–1.23)	0.593	1.06 (0.66–168)	0.811
Chronic kidney disease						
No	1	—	1	—	1	—
Yes	3.16 (2.45–4.07)	**<0.001**	3.02 (2.25–4.05)	**<0.001**	3.57 (2.11–6.03)	**<0.001**
Alcohol consumption						
No	1	—	1	—	1	—
Yes	1.08 (0.80–1.45)	0.592	1.17 (0.86–1.58)	0.332	1.84 (0.22–5.28)	0.988
Ischemic heart disease						
No	1	—	1	—	1	—
Yes	1.45 (1.04–2.02)	**0.032**	1.49 (1.03–2.11)	**0.034**	1.69 (0.68–4.20)	0.293
COPD						
No	1	—	1	—	1	—
Yes	1.76 (1.31–2.37)	**<0.001**	1.91 (1.40–2.60)	**<0.001**	1.63 (0.42–642)	0.443
Heart failure						
No	1	—	1	—	1	—
Yes	3.85 (2.94–5.04)	**<0.001**	4.33 (3.19–5.89)	**<0.001**	2.42 (1.39–4.21)	**0.003**
Leukemia						
No	1	—	1	—	1	—
Yes	3.09 (1.41–6.78)	**0.007**	2.40 (0.92–6.33)	0.106	5.18 (1.14–22.42)	**0.038**
Lymphoma						
No	1	—	1	—	1	—
Yes	2.09 (1.00–4.38)	0.061	2.32 (0.96–5.64)	0.078	1.43 (0.37–5.49)	0.706
Cirrhosis						
No	1	—	1	—	1	—
Yes	1.68 (0.78–3.58)	0.17	1.88 (0.88–4.03)	0.102	NC	1.00
Hepatitis C						
No	1	—	1	—	1	—
Yes	1.63 (0.86–3.04)	0.138	1.79 (0.93–3.45)	0.531	0.42 (0.10–8.54	1.00
HIV						
No	1	—	1	—	1	—
Yes	1.03 (0.30–3.61)	1.00	0.39 (0.05–3.04)	0.213	7.69 (0.69–85.23)	0.112
Transplant						
No	1	—	1	—	1	—
Yes	1.92 (0.80–4.60)	0.174	2.32 (0.96–0.56)	0.078	NA	1.00
Sepsis						
No	1	—	1	—	1	—
Yes	5.48 (4.54–6.61)	**<0.001**	5.59 (4.53–6.91)	**<0.001**	6.02 (3.32–7.52)	**<0.001**
Acute kidney injury						
No	1	—	1	—	1	—
Yes	6.37 (5.23–7.55)	**<0.001**	6.25 (5.03–7.80)	**<0.001**	6.84 (4.41–10.71)	**<0.001**
Pulmonary embolism						
No	1	—	1	—	1	—
Yes	2.76 (1.28–5.97)	**0.018**	3.02 (1.27–7.11)	**0.016**	1.98 (0.34–10.53)	0.610
ICU admission						
No	1	—	1	—	1	—
Yes	3.12 (2.63–3.74)	**<0.001**	3.19 (2.60–3.92)	**<0.001**	3.00 (2.02–4.47)	**<0.001**

^a^ Significant *p* values (<0.05) are in bold. Abbreviations: CI, confidence interval; COPD, chronic obstructive pulmonary disease; HIV, human immunodeficiency virus; ICU, intensive care unit; *N*, sample size (number of admissions); OR, odds ratio; NA: not available.

## Data Availability

The data presented in this study are available upon request from the corresponding author. The database is provided by the Spanish Health Ministry and is intended for research or administrative purposes; therefore, it is not freely available to the public.

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
