# Peer review of "National Study of Fournier Gangrene in Spain (2016–2021): Gender/Sex Differences in Mortality and Risks"

_medicina, 2024, doi:10.3390/medicina60101600_

Round 1

Reviewer 1 Report

Comments and Suggestions for Authors

One of the rare studies investigating Fournier Gangrene (FG). However, some points need to be explained.

Only diagnosed patients were evaluated in the study, and differences were found to be significant according to age and gender.

It would be appropriate to give the changes in the rates according to years in the form of standardized rates with one of the standardization methods (direct standardization is more appropriate), both in mortality and morbidity.

To test the fit of the model, the area under the receiver operating characteristic curve (AUC) was calculated.

There is an R2 statistic used to measure the strength of the relationship between dependent and independent variables in logistic regression. These statistics are the CoxSnell R2 statistic and the Nagelkerke R2 statistic.

What were the Cox-Snell R2 and Nagelkerle R2 values?

Author Response

Comments 1: It would be appropriate to give the changes in the rates according to years in the form of standardized rates with one of the standardization methods (direct standardization is more appropriate), both in mortality and morbidity.

Response 1: According to your comment, we have calculated  the standardized hospitalization rates by direct methods.

For instance we have:

Include in material and methods the following sentence Hospitalization and mortality rates were adjusted by age using the direct method of standardization, with Eurostat 2013 data as reference population

IN the results we have modified the follow sentence. The crude and standardized hospitalization rates were approximately five times higher in men than in women (Table 2).

And the able 2 has been modified

Men

Women

Total

N

RHa

RHb

N

RHa

RHb

N

RHa

RHb

Crude Rates

2016

366

0.17

1.6

91

0.04

0.4

457

0.10

1.0

2017

386

0.18

1.7

91

0.04

0.4

477

0.10

1.0

2018

510

0.23

2.2

121

0.05

0-5

631

0.14

1.4

2019

492

0.22

2.1

101

0.04

0.4

593

0.13

1.3

2020

581

0.30

2.5

124

0.06

0.5

705

0.17

1.5

2021

670

0-32

2.9

111

0.05

0.5

781

0.18

1.6

Mean

490

0.23

2.7

106

0.05

0.4

596

0.14

1.7

Standarized rates

2016

-

0.16

1.60

-

0.04

0.40

-

0.10

0.99

2017

-

0.17

1.70

-

0.04

0.40

-

0.11

1.03

2018

-

0.23

2.23

-

0.05

0.51

-

0.14

1.35

2019

-

0.22

2.14

-

0.04

0.42

-

0.13

1.26

2020

-

0.30

2.50

-

0.06

0.51

-

0.18

1.49

2021

-

0.32

2.90

-

0.05

0.46

-

0.18

1.64

* Adjusted by age using the direct method of standardization

Moreover we have include a new sub-section in results including crude and standardized rate of death, and also a new table. For instance, we have changed the number of the rest of variables

Crude and Standarized Rate  of mortality  

The mortality rate was higher in women than in men (20.9% vs 15.1% with OR 1.31, 95% CI 1.09-1.56).  The crude and standardized rate of mortality is in table 4.

Table 4. Episodes, Crude and Standarized Rate* of mortality per 1,000 Spanish cases of death and per 100,000 Inhabitants (Men, Women, and Both).

Men

Women

Total

N

RHa

RHb

N

RHa

RHb

N

RHa

RHb

Crude rates

2016

67

0.22

0.29

19

0.08

0.08

86

0.15

0.18

2017

67

0.31

0.29

15

0.07

0.06

82

0.19

0.18

2018

65

0.30

0.28

25

0.19

0.10

90

0.21

0.19

2019

66

0.31

0.28

23

0.11

0.10

89

0.21

0.19

2020

87

0.35

0.37

25

0.10

0.10

112

0.23

0.23

2021

103

0.44

0.44

26

0.12

0.10

129

0.29

0.27

Mean

75

0.31

0.33

22

0.010

0.09

97

0.21

0.21

Standardized rates*

2016

-

0.4

0.33

0.33

0.08

0.43

0.19

2017

-

0.50

0.32

0.14

0.05

0.38

0.17

2018

-

0.41

0.31

0.24

0.09

0.35

0.19

2019

-

0.38

0.31

0.17

0.08

0.31

0.18

2020

-

0.63

0.40

0.48

0.10

0.58

0.24

2021

-

0.58

0.48

0.23

0.09

0.46

0.27

Mean

-

0.50

0.36

0,27

0.08

0.42

0. 21

* Adjusted by age using the direct method of standardisation

Abbreviations: N, sample size (number of admissions); RH, rate of hospitalizations.

aNumber of hospitalizations for Fournier gangrene per 1,000 all od death during admission per year.

  bNumber of hospitalizations for Fournier gangrene per 100,000 inhabitants per year

Comments 2: To test the fit of the model, the area under the receiver operating characteristic curve (AUC) was calculated.

Response 2:  It was performed

Comments 3: There is an R2 statistic used to measure the strength of the relationship between dependent and independent variables in logistic regression. These statistics are the CoxSnell R2 statistic and the Nagelkerke R2 statistic.

What were the Cox-Snell R2 and Nagelkerle R2 values?

Response3 : Thanks for you comment.

We have included in the Data Analysis su-section that:… The CoxSnell R2 statistic and the Nagelkerke R2 statistic were use to used to measure the strength of the relationship between dependent (mortality) and independent variables.

And in the Risk Factors for Mortality Among All Patients sub-section (results section: The model has a Cox-Snell R2 value of 0.226 and Nagelkerle R2 value of 0.352

And in the Risk Factors for Mortality Among Men and Women Sub-section (results section).

For men… The model has a Cox-Snell R2 value of 0.184 and Nagelkerle R2 value of 0.321.

For women The model has a Cox-Snell R2 value of 0.213 and Nagelkerle R2 value of 0.333

Reviewer 2 Report

Comments and Suggestions for Authors

The article is well written,and is based on a huge database.

I have only  minor comments.

Please explain why did you utilised the Mann-Whitney U test,a nonparametric test for continuous variables,and not a parametric test,considering the large cohort number?

I recommend to replace citations 14 and 22,and insert citations about the influence of age in FG.

Author Response

Comments 1: Please explain why did you utilised the Mann-Whitney U test,a nonparametric test for continuous variables, and not a parametric test,considering the large cohort number?

Response 1: Thanks for you comment, we want to say that the for continuous variables, we calculated distributed according to the Kolmogorov-Smirnov test and the value was < 0.05 (significant and no follow a normal distribution) for instance we use medians and interquartile ranges (IQRs) and there  were analyzed using the Mann-Whitney U test.

We have added for explain that… continuous variables were analyzed using the Mann-Whitney U test because the distribution was no normal (p value < 0.005 in Kolmogorov-Smirnov test).

Comments 2: I recommend to replace citations 14 and 22, and insert citations about the influence of age in FG.

Response 2: Thanks for you comment. According you suggestion, we have removed in the text although age is generally a mortality risk factor in people with infectious diseases, such as COVID-19 and flu14,23. And the reference 14 and 23.

And we have included citations about the age in FG … as other studies.22-23

 Furr J, Watts T, Street R, et al. Contemporary trends in the inpatient management of Fournier’s gangrene: predictors of length of stay and mortality based on population-based sample. Urology 2017; 102: 79–84

Kim SY, Dupree JM, Le BV, et al. A contemporary analysis of Fournier gangrene using the National Surgical Quality Improvement Program. Urology 2015; 85: 1052–1057.

Reviewer 3 Report

Comments and Suggestions for Authors

The presented research is relevant and the results obtained and processed statistically are interesting. One of the most interesting findings revealed by the authors is that women have a higher risk of mortality, despite men being more likely to become sick. The authors have honestly pointed out the limitations of their study in the discussion sectionand these limitations cannot be disputed. There are no significant comments to be made. Given the importance of the article.

Author Response

Comments 1: The presented research is relevant, and the results obtained and processed statistically are interesting. One of the most interesting findings revealed by the authors is that women have a higher risk of mortality, despite men being more likely to become sick. The authors have honestly pointed out the limitations of their study in the discussion section, and these limitations cannot be disputed. There are no significant comments to be made. Given the importance of the article.

Response 1: Thanks for your comments.
